# Is Scale All You Need For Anomaly Detection?

## Abstract

Scaling up neural representations has led to an unprecedented boost in anomaly detection methods' performance. This paper tackles the question: can we solve anomaly detection with arbitrary accuracy by continuing to scale up neural representations? We begin by highlighting that overly expressive representations are often unable to detect even simple anomalies when evaluated beyond well-studied object-centric datasets. We explain this phenomenon by introducing a theoretical toy model for anomaly detection performance. The model provides evidence for a no-free-lunch theorem in anomaly detection stating that increasing representation expressivity will eventually result in performance degradation. To break this deadlock, it is necessary to provide guidance to focus the representation on the attributes relevant to the anomalies of interest. We conducted an extensive empirical investigation demonstrating that state-of-the-art representations often suffer from over-expressivity, failing to detect many types of anomalies in practical settings. Our paper underscores that achieving breakthroughs in anomaly detection requires more than just scale; it requires making informed assumptions about the nature of the anomalies.

## 1 Introduction

Anomaly detection, identifying patterns that deviate significantly from the norm, is a crucial problem for science and industry. It has many applications ranging from identifying fraudulent transactions in financial systems to scientific phenomena that deviate from an existing paradigm (e.g. supernovae, quantum physics). Different from supervised learning, which automates known processes, anomaly detection seeks to discover the unknown. This is a more challenging task but with potentially very significant benefits. New scientific and industrial discoveries are crucial for tackling the challenges facing humanity and may benefit from better anomaly detection capabilities. This makes advancing anomaly detection methods a high-priority task.

The introduction of deep networks has resulted in significant improvements in the state-of-the-art performance on anomaly detection benchmarks, for images (Reiss et al., 2021), videos (Georgescu et al., 2021), time-series (Qiu et al., 2021) and other modalities. This begs the question:

*Can scaling the current approaches lead to universal anomaly detection?*

We begin our investigation by conducting a preliminary experiment to examine the impact of increasing representation expressivity on anomaly detection performance. As we gradually enhance the expressivity of the representation, we reach a critical point where irrelevant attributes are incorporated into the representation, which does not improve detection performance. To better understand this phenomenon, we present a model that shows how discovering some anomalies necessarily comes at the expense of missing others. For example, consider an explorer inspecting stones in a mine. Many of the stones are anomalous in different ways; one stone is the largest, another stone can be the hardest, and a third is the softest. Intuitively, if we wish to order the stones from the most anomalous to the least anomalous, we have a trade-off; the more attributes we may wish to account for, the less sensitive the detector becomes to anomalies in each individual attribute.

We propose a simple mathematical model for the trade-off demonstrated by the above motivating example. Let the sample (e.g., a stone) be described by a set of $d$ attributes (e.g., hardness, size, weight). Let us further assume that the first attribute (e.g., hardness) is the important anomaly and has an unusual value, and all other attributes are normal (e.g., a very hard stone with normal weight and size). This makes $d-1$ attributes irrelevant for the purposes of detecting the important anomaly.

We show that increasing the number of (irrelevant) attributes makes the method significantly less sensitive. Specifically, for a fixed false positive detection rate (FPR), the true positive rate (TPR) of the anomaly classification converges to the FPR as we increase $d$ at a rate of $\frac{1}{\sqrt{d}}$. The above model reveals a simple bias-variance trade-off: On one hand, the representation must include the attribute relevant to separating normal and anomalous data among its $d$ attributes. As this attribute is not known in advance, this encourages large representations that are expressive of all possible attributes. On the other hand, including irrelevant attributes reduces the sensitivity of the method and leads to a convergence towards a random classifier, encouraging minimal representations.

Navigating the above trade-off requires using a prior over the attributes most likely to lead to important anomalies. There is, however, an inherent conflict between priors and anomaly discovery; anomalies are unexpected and counter-intuitive and are likely to defy priors. Although most existing methods do not explicitly acknowledge using a prior, we show that the leading approaches are strongly biased toward anomalies within a particular set of attributes (mostly novel object category or deformations). Although this is advantageous in several cases, including current academic benchmarks, we show that these implicit priors fail when they are not matched with the desired anomaly. Similar conclusions follow for the out-of-distribution (OOD) detection and outlier exposure (OE) settings.
Our main contributions are:

1. Identifying over-expressivity as a bottleneck in scaling deep anomaly detection.
2. Proposing a toy model for quantifying the bias-variance trade-off resulting in a "no free lunch" (NFL) principle for anomaly detection.
3. Empirical analysis demonstrating and quantifying how current methods are affected by the above NFL principle.

## 2 RELATED WORK

Anomaly detection has been extensively studied in the machine learning community, and a wide range of methods have been proposed over the years. Here, we provide an overview of relevant works, focusing on representation learning in anomaly detection, challenges associated with over-expressivity, and utilizing guidance to detect anomalies.

**Representation learning in anomaly detection.** Anomaly detection has traditionally relied on classical approaches such as density estimation (Eskin et al., 2002; Latecki et al., 2007) or reconstruction (Jolliffe, 2011). However, deep learning has revolutionized anomaly detection by incorporating deep representations into these classical methods (Mathieu et al., 2016; Zhang et al., 2016; Larsson et al., 2016; Noroozi & Favaro, 2016). Deep representation learning for anomaly detection often involves self-supervised learning techniques. One common approach is the use of autoencoders (Ruff et al., 2018), which learn to reconstruct input data by compressing it into a low-dimensional representation and then decoding it back to its original form. Another self-supervised method is rotation classification (Golan & El-Yaniv, 2018; Hendrycks et al., 2019), where the model learns to classify the correct rotation of an input image, forcing it to capture useful visual features. Contrastive learning methods (Tack et al., 2020; Sohn et al., 2020; Wang et al., 2022; Zou et al., 2022; Yao et al., 2023) have also been employed. These methods aim to learn representations that maximize the similarity between augmented versions of the same image while minimizing similarity to other images. Another representation learning paradigm for anomaly detection is to combine pretrained representations with anomaly scoring functions (Perera & Patel, 2019; Reiss et al., 2021; Roth et al., 2022; Cohen & Avidan, 2022; Liu et al., 2023; Zhang et al., 2023; Jeong et al., 2023). This involves using representations pretrained on large-scale external dataset, such as ImageNet classification, and then fine-tuning them for the specific anomaly detection task using the provided normal samples in the training set. These methods achieve high performance in academic anomaly detection datasets.

**Over-expressivity.** The concept of over-expressivity has been discussed in the general machine learning literature (Shalev-Shwartz & Ben-David, 2014), but its specific implications for anomaly detection have been relatively overlooked. Although deep learning approaches to anomaly detection have achieved impressive performance on academic anomaly detection datasets, recent studies have revealed limitations in detecting anomalies beyond well-studied object-centric datasets. It has been observed (Doorenbos et al., 2022) that even highly expressive representations fail to identify simple anomalies when the anomalies lie in unexplored attribute spaces. Although Doorenbos et al. (2022)

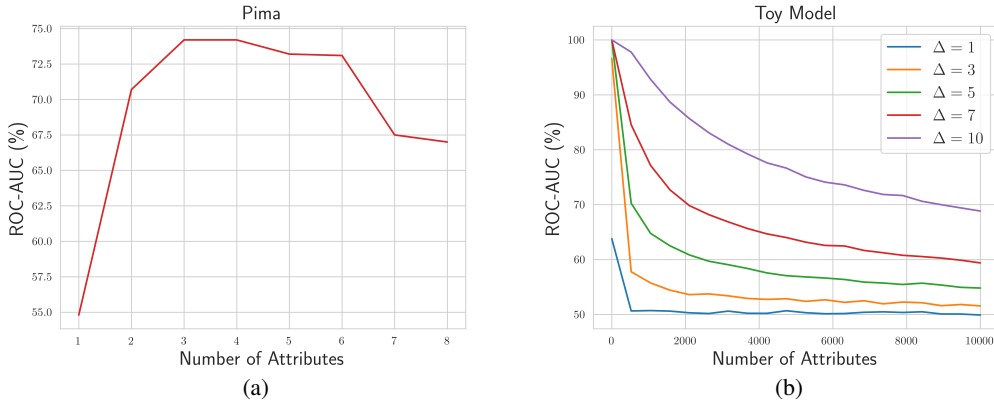

Figure 1: (*a*) Anomaly detection performance on the Pima dataset. Once the relevant attributes are included, the representation expressivity increases. (*b*) An illustration of our model. As the number of attributes increases, the classifier gets closer to random prediction.

suggested that using the Mahalanobis distance (essentially a Gaussian density model) might mitigate this issue, our analysis shows that problems remain even under such models.

**Guidance in anomaly detection.** Guidance plays a crucial role in anomaly detection, as prior knowledge is often required to effectively identify unexpected patterns. Approaches such as PANDA (Reiss et al., 2021) and PatchCore (Roth et al., 2022) utilize pre-trained representations, incorporating implicit priors from ImageNet or CLIP (Jeong et al., 2023; Reiss & Hoshen, 2022) to enhance anomaly detection. Out-of-distribution (OOD) detection methods (Hendrycks et al., 2020; Fort et al., 2021; Galil et al., 2023) assume anomalies have unusual values in pre-specified attributes, whereas outlier exposure methods (Hendrycks et al., 2018; Mirzaei et al., 2022) rely on external datasets resembling anomalies. Other methods (Zhang & Ranganath, 2023; Cohen et al., 2022) allow users to specify irrelevant attributes. Similarly, self-supervised methods (Hendrycks et al., 2019; Tack et al., 2020; Sohn et al., 2020; Schlüter et al., 2022; Zou et al., 2022; Wang et al., 2022; Yao et al., 2023) remove nuisance factors from the representation using augmentation. In this paper, we analyze guidance through our developed framework, exploring its impact on the trade-off between representation expressivity and anomaly detection performance (Sec. 5.3).

## 3 A MOTIVATING EXAMPLE: TRADE-OFF BETWEEN EXPRESSIVITY AND REPRESENTATION SUFFICIENCY

We aim to elucidate the trade-off between over-expressivity and the sufficiency of representations in anomaly detection. In this section, we motivate this research by considering a simple, tabular anomaly detection task. The established Pima dataset (from ODDS Rayana (2016)), contains biomarker attributes for a set of patients, including pregnancies, glucose levels, blood diabetes pedigree, insulin, and age. Each patient is also labeled as healthy or having diabetes. The Pima dataset is converted to anomaly detection by considering healthy patients as normal data. At training time, an anomaly detector is trained on a training set containing only healthy patients. At test time, a new patient is evaluated and the model is required to predict whether the patient is normal (healthy) or anomalous. The evaluation metric is the area under the ROC curve (ROC-AUC) between the model predictions and the ground truth healthy/diabetes labels. For this illustrative example, we use a $k$NN anomaly detector. Despite its simplicity, this simple detector achieves state-of-the-art results on this dataset.

The expressivity of the representation is critical to our approach. We model expressivity as the number of included attributes, $d$. We gradually increase the expressivity of the representation $d$ by starting from a single attribute ($d = 1$) and adding features (by relevancy) to the representation until it contains all the provided attributes ($d = 8$). The results are provided in Fig. 1a. The graph shows that anomaly detection performance increases as the number of attributes increases from $d = 1$ to $d = 4$. However, with every additional attribute that is added, results are degraded, with the lowest level at $d = 8$. We accredit the observed phenomenon to the sufficiency and over-expressivity of the representation. As four out of the eight attributes are effective for separating patients with and without diabetes,

performance initially increases. However, the remaining four attributes are a nuisance for the anomaly detection task, making the representation over-expressive. When including irrelevant attributes in the representation, the density estimator becomes less sensitive to the anomalies of interest. The presence of non-informative attributes reduces the discriminative power of the representation. It demonstrates that blindly increasing representation expressivity does not guarantee improved detection performance. Instead, it is crucial to navigate the trade-off between representation complexity and sensitivity to anomalies within specific attributes. This requires priors on the nature of the anomalies, which itself requires some form of guidance that can be implicit or explicit.

## 4 BIAS-VARIANCE TRADEOFF

In this section, we use a simple toy model to demonstrate the tradeoff between sufficiency and over-expressivity. In this model, the dimensionality of the representation is precisely correlated to the number of included attributes, i.e., a change in dimensionality directly affects the number of attributes. The model shows that increasing the number of attributes used by an anomaly detector will eventually reduce the anomaly detection capabilities. We show that as the number of dimensions attributes $d$ tends to infinity, the sensitivity (true positive rate) approaches the false positive rate. As random detectors also achieve sensitivity equal to the false positive rate, this constitutes a complete failure of the detector. We focus on anomaly detection by density estimation as the leading methods for both image anomaly detection and segmentation tasks are based on density estimation (Reiss et al., 2021; Sohn et al., 2020; Roth et al., 2022; Yao et al., 2023). Our theorem is illustrated in Fig. 1b.

### 4.1 A TOY MODEL

We model the normal data by a multivariate Gaussian distribution of $d$ dimensions with an identity covariance matrix. We further assume that the anomalous data distribution is identical to the normal data except for a shift in the first axis. There is no loss of generality as the distributions are spherically symmetric, and therefore any offset between the means of the distributions can be rotated to align with the first axis. We show that following the result holds:

**Theorem 1.** *Let the population distribution of samples (D) be given by:*

$$D = (1 - \pi) \cdot \mathcal{N}(0, I) + \pi \cdot \mathcal{N}(\Delta \cdot e_1, I) \tag{1}$$

*where $\pi$ is a Bernoulli distribution describing the probability of a given sample to be anomalous. We assume an unknown ground truth anomaly labeling function $l : D \to \{0, 1\}$. We compute the anomaly score for test samples using their likelihood with respect to the normal data distribution: $s_d(x) = \frac{e^{-\frac{1}{2}||x||^2}}{\sqrt{2\pi}^d}$. Samples are classified as anomalous if their likelihood $s_d(x)$ is lower than the threshold $t_d$. We prove the following statements hold:*

1. *The true positive rate converges to the false positive rate as the dimension $d$ tends to infinity.*

2. *For large values of $d$, the difference between the TPR and the FPR decays as $\frac{1}{\sqrt{d}}$.*

*Proof Sketch of Theorem 1* We begin with the following lemma.

**Lemma 1.** *Let $t_d(y) \equiv \frac{e^{-\frac{1}{2}(y\sqrt{2d}+d)}}{\sqrt{2\pi}^d}$, where $y \in \mathbb{R}$ parameterized the sensitivity of the method. For a sufficiently large $d$ for the central limit theorem to hold, the false positive rate of the method converges to a constant.*

The proof of Lemma. 1 is provided in App. A. Now, having set the threshold as $t_d(y)$, the false positive rate is fixed; we compute the dependence of the true positive rate on the dimension $d$. To calculate the true positive rate, we need to compute the distribution of squared norms $||x||^2$ of anomalous examples. By Horgan & Murphy (2013), we have that for large $d$ the distribution is:

$$||x||^2 \sim \mathcal{N}(2d + \Delta^2, 4\Delta^2 + 2d)$$

In App. B, we derive an expression for the true positive rate:

$$TPR(t_d) = Pr(||x||^2 > y\sqrt{2d} + d \mid l(x) = 1) = \dots$$

$$= \int_{y\sqrt{2d}+d}^{\infty} \frac{1}{\sqrt{2\pi \cdot (4\Delta^2 + 2d)}} \cdot e^{-\frac{1}{2} \cdot (\frac{y' - (d+\Delta^2)}{\sqrt{4\Delta^2+2d}})^2} dy' \underset{\lim_{d\to\infty}}{\approx} FPR(t_d) + \Delta^2 \cdot \frac{1}{\sqrt{2\pi \cdot 2d}} \cdot e^{-\frac{1}{2} \cdot y^2}$$

We see that the true positive rate decomposes into two terms: the false positive rate and another factor converging to 0 at a rate of $\frac{1}{\sqrt{d}}$. As equal true positive and false positive rates indicate a random detector, the result describes convergence toward random predictions.

## 4.2 No Free Lunch

Our theoretical analysis reveals a fundamental trade-off between representation sufficiency and over-expressivity in anomaly detection. Although it is tempting to believe that increasing the expressivity of representations will lead to continuous improvements in anomaly detection performance, we provide evidence to the contrary. Building upon these findings, we introduce the concept of the *"no free lunch"* principle in anomaly detection; including an ever-increasing number of attributes reduces our ability to detect anomalies in each of them. Therefore, trying to build a detector that would catch all types of anomalies is impossible, and detection must rely on some prior assumption regarding the nature of the anomalies we look for. In other words, there is no universal anomaly detection method that excels in all scenarios without explicit guidance.

This finding prompts us to explore the practical implications of over-expressivity in state-of-the-art methods. Thus, in what follows, we conduct an extensive empirical investigation to demonstrate how current anomaly detection methods are affected by this "no free lunch" principle. By evaluating these methods across diverse datasets and anomaly types, we aim to showcase the real-world implications of over-expressive representations and shed light on the limitations of existing approaches.

## 5 Quantifying The Effect of Over-Expressivity

We investigated the impact of over-expressivity and the limitations of existing anomaly detection methods. We selected a range of datasets to represent a wide range of anomalies and test whether anomaly detection approaches generalize beyond novel object class anomalies.

**Datasets.** The following datasets were used in our analysis: *Cars3D* (Reed et al., 2015): This dataset consists of 17,568 3D car models with variations in elevation, rotation (azimuth), and car model (object). *RaFD* (Langner et al., 2010): The Radboud Faces Database (RaFD) contains 8,040 facial images of different individuals expressing various emotions. Each image is annotated with emotional attributes such as happiness, anger, sadness, etc as well as person identity and head pose (angle). *CelebA* (Liu et al., 2015): The CelebA dataset consists of celebrity face images with annotated attributes. In Fig. 2 we present a visualization of the dataset samples and the attributes we used for our evaluations. For each dataset, we split the data into training and testing sets with an 85:15 ratio. We trained the anomaly detection models on the training set and evaluated their performance on the testing set. Full training and implementation details are in App. C.

**Baselines.** We conducted evaluations on prominent self-supervised and pretrained feature adaptation anomaly detection methods, along with an out-of-distribution (OOD) method (Fort et al., 2021) that incorporates guidance on the relevant attribute. The baselines we considered include DN2 with unadapted ImageNet pretrained ResNet features, PANDA (Reiss et al., 2021), MSAD (Reiss & Hoshen, 2023), pretrained DINO (Caron et al., 2021; Reiss et al., 2023), and CSI (Tack et al., 2020). To explore the impact of using guidance, we provided attribute values during the training phase for the OOD method. Specifically, we followed the state-of-the-art OOD method (Fort et al., 2021) which finetune a pretrained backbone using multi-class classification objective (provided by attribute values guidance information). It is important to note that the model was trained exclusively on normal data, like the other baselines. All methods are applied with their default hyperparameters as given in their official implementations. Unless otherwise specified, they use the same ResNet18 (He et al., 2016) backbone architecture (except DINO, which utilizes a ResNet50 architecture).

## 5.1 Experimental Setting

To evaluate the efficacy of each method, we conducted experiments among many datasets and settings: multi-value normal data and one-class classification. This comprehensive approach allows us to capture the overall performance trends and provide a comprehensive analysis. We report the average performance (ROC-AUC metric) obtained in each evaluation, providing a robust assessment of the models' capabilities. We used a standard $k$-nearest neighbors ($k$NN) density estimator for anomaly

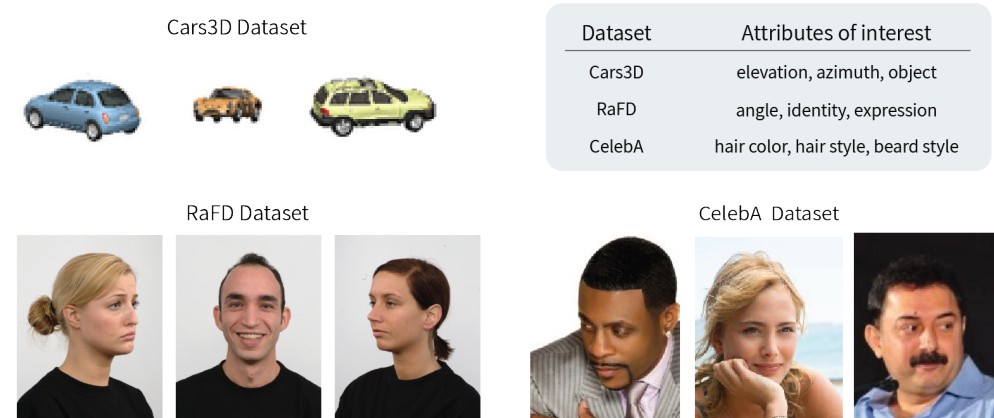

Figure 2: Representative images of the different datasets, and their attributes of interest.

scoring. To assess the over-expressivity of our baselines, we adopted a supervised linear probing approach on the *representation already learned* from the anomaly detection task. This involved augmenting the training set with anomalous samples and training a linear classifier on the baselines' *representations* to differentiate between normal and anomalous instances. The resulting logits from the linear probing experiment were used as the anomaly scores. The same test sets were used for both the $k$NN density estimator and linear probing anomaly scoring. In each experiment, we evaluated the performance of the anomaly detection models across the first three values of each attribute within the given dataset. The implementation details can be found in App. C.

**Single-value and multi-value classification settings.** We examine both the usual single-value setting, and the more challenging multi-value classification settings. With the multi-value setting, we are able to directly compare the unguided anomaly detection baselines with the OOD baseline that use attribute-specific guidance (see Sec. 5.3). In the single-value setting, the normal data consists of a single semantic class (one-class classification), and no labels are provided. For each experiment, we designated a single value as *normal* while considering all other values as *anomalous*. We further assessed the performance of the methods in the multi-value setting. In this setting, the normal data consists of multiple semantic classes, and no labels are provided. For each experiment conducted in this setting, we designated a single value as *anomalous* while considering all other values as *normal*. In both settings, the representations were trained solely on the normal samples.

## 5.2 MAIN RESULTS

We present the evaluation results for the single-value setting in Tab. 1. Our findings reveal inconsistencies in anomaly detection across different attributes of interest. Surprisingly, even in the case of the synthetic Cars3D dataset, where we initially expected the baselines to perform very well, we observe low accuracy in detecting elevation and azimuth anomalies. We found a notable difference between $k$NN and linear probing performance for each method. This gap emphasizes the over-expressivity of the representations, as the $k$NN estimator becomes less sensitive to the anomalies of interest. Linear classifier achieves higher anomaly detection accuracy as it can learn to focus on the relevant attributes. The results on Cars3D underscore the over-expressivity of the representation and indicate that very expressive representations fail to detect simple anomalies when evaluated beyond the well-studied object-centric datasets. In the RaFD dataset, our baselines demonstrate strong performance in detecting anomalies related to the angle and identity attributes. However, they struggle when confronted with anomalies associated with the expression attribute. Similarly, in CelebA, we encounter an even more challenging scenario where all baselines exhibit poor performance across all attributes. In some cases, the limited effectiveness of the linear probing approach indicates the representations are not sufficiently expressive, as they fail to sufficiently encode the relevant attribute information. Consequently, the representation cannot effectively differentiate between normal and anomalous data.

Tab. 2 presents the results of the multi-value setting, where we conducted additional evaluations of our baselines and observed similar trends to those in the single-value setting. However, the multi-value setting provided an opportunity to investigate the OOD baseline that incorporates attribute-specific

Table 1: Single-value anomaly detection performance (mean ROC-AUC %). *k*NN best in **bold black** and linear probing best in **bold blue**.

|  | Method | CSI | | DN2 | | PANDA | | MSAD | | DINO | |
|---|---|---|---|---|---|---|---|---|---|---|---|
|  | Relevant attr. | *k*NN | Lin. | *k*NN | Lin. | *k*NN | Lin. | *k*NN | Lin. | *k*NN | Lin. |
| Cars3D | Elevation | 69.8 | 88.2 | 79.5 | 88.6 | **80.2** | 88.4 | 79.5 | **89.0** | 77.4 | 83.7 |
| | Azimuth | 73.1 | 93.6 | 89.2 | **98.0** | 90.4 | 97.9 | 88.6 | **98.0** | **91.1** | 97.3 |
| | Object | 91.7 | 86.5 | 99.7 | **99.9** | 99.7 | **99.9** | 99.7 | **99.9** | **99.9** | **99.9** |
| | Average | 78.2 | 89.4 | 89.5 | 95.5 | **90.1** | 95.5 | 89.3 | **95.7** | 89.5 | 93.5 |
| RaFD | Angle | 89.2 | 95.2 | 98.2 | **100.** | 98.9 | **100.** | 99.5 | 99.9 | **99.9** | **100.** |
| | Identity | 83.2 | 85.2 | 98.2 | 99.9 | 98.2 | 99.9 | **100.** | **100.** | **100.** | **100.** |
| | Expression | 53.8 | 66.0 | 57.2 | 82.3 | 57.2 | 81.1 | 56.7 | 83.1 | **60.0** | **89.2** |
| | Average | 75.4 | 82.1 | 84.5 | 94.1 | 84.8 | 93.7 | 85.4 | 94.3 | **86.7** | **96.1** |
| CelebA | Hair color | 50.2 | 80.9 | 56.9 | 84.5 | 57.0 | 82.0 | **59.7** | 85.3 | 58.5 | **87.1** |
| | Hair style | 51.1 | 79.5 | 54.4 | 81.6 | 54.5 | 79.9 | **56.6** | 82.4 | 55.3 | **84.6** |
| | Beard style | 50.8 | 87.8 | 57.6 | 91.5 | 58.9 | 90.8 | **62.6** | 91.1 | 60.6 | **94.2** |
| | Average | 50.7 | 82.7 | 56.3 | 85.9 | 56.8 | 84.2 | **59.6** | 86.3 | 58.1 | **88.7** |

guidance. Our results strongly support the notion that prior knowledge of the relevant attributes significantly aids in discriminating anomalies. Although the performance is still imperfect, OOD outperformed the other baselines. Moreover, the linear probing results demonstrate nearly perfect performance for all attributes in Cars3D and RaFD, surpassing other baselines by a large margin. This highlights the over-expressivity of the representation, particularly in the context of the OOD baseline, and underscores the importance of incorporating guidance regarding the relevant attributes.

## 5.3 ANALYSIS OF GUIDANCE IN LEADING ANOMALY DETECTION ALGORITHMS

Having established that representations must be sufficient but not over-expressive, it is clear that guidance must be used to construct appropriate representations. While anomaly detection methods strive to use minimal guidance, nearly all deep anomaly detection methods use some form or another. Here, we analyze the top-performing anomaly detection paradigms through the lens of guidance for reducing representation over-expressivity while preserving the relevant information.

**Self-supervised learning (SSL) methods.** SSL methods (Golan & El-Yaniv, 2018; Tack et al., 2020; Sohn et al., 2020; Yao et al., 2023) have gained popularity in anomaly detection due to their ability to reduce labeling burden and impose weaker priors on expected anomalies. However, SSL methods still provide quite strong priors on the expected anomalies. Contrastive methods, which are the top-performing SSL methods, require representations to be invariant to well-designed augmentations (e.g. horizontal reflections or translations). While this reduces the number of degrees of freedom in the representation $d$, reducing over-expressivity, it poses a challenge when the anomaly lies within the invariant factors, such as an unusual horizontal flip or image translation. Many SSL methods also use negative augmentations to guide the representation to specified relevant attributes, e.g., rotations in contrastive methods (Tack et al., 2020; Sohn et al., 2020) or generative models in anomaly segmentation (e.g. Li et al. (2021); Zou et al. (2022)). While this increases representation sufficiency, the choice of augmentations must align with the characteristics of the (unknown) anomalies.

**Pretrained representations.** Recently, anomaly detection methods based on pretrained representations, particularly those based on ImageNet or CLIP, have emerged as dominant approaches in image anomaly detection and segmentation benchmarks (Reiss et al., 2021; Roth et al., 2022). These pretrained representations have significantly increased the expressivity of the models, as shown in Tab. 1. By leveraging large-scale pretraining on massive datasets, these representations capture a wide range of visual patterns and features, enhancing their potential to be sufficient for anomaly detection tasks. However, it is important to note that the increase in expressivity is not uniform across all aspects of anomalies. While pretrained representations can help reduce over-expressivity and improve performance on certain types of anomalies, they also introduce the risk of insufficient representations when the guidance is not aligned with the anomalies of interest. Various pretrained methods focus on different attributes, such as detecting anomalous objects at coarse granularity versus

Table 2: Multi-value anomaly detection performance (mean ROC-AUC %). $k$NN best in **bold black** and linear probing best in **bold blue**.

| | Method | CSI | | DN2 | | PANDA | | MSAD | | OOD | |
|---|---|---|---|---|---|---|---|---|---|---|---|
| | Relevant attr. | $k$NN | Lin. | $k$NN | Lin. | $k$NN | Lin. | $k$NN | Lin. | $k$NN | Lin. |
| Cars3D | Elevation | 67.6 | 99.1 | 58.5 | 88.6 | 51.8 | 87.6 | 59.2 | 89.3 | **69.1** | **99.2** |
| | Azimuth | 42.2 | 97.4 | 82.4 | 98.0 | 79.8 | 98.5 | 81.8 | 98.4 | **84.1** | **99.9** |
| | Object | 75.0 | 99.7 | 87.9 | **99.9** | 78.1 | 98.6 | 85.1 | **99.9** | **96.8** | **99.9** |
| | Average | 61.6 | 98.7 | 76.2 | 95.5 | 69.9 | 94.9 | 75.3 | 95.9 | **83.3** | **99.7** |
| RaFD | Angle | 88.0 | 98.5 | 97.7 | **100.** | 88.5 | **100.** | 98.5 | **100.** | **100.** | **100.** |
| | Identity | 66.9 | 85.4 | 97.5 | 99.9 | 93.4 | 99.7 | 99.3 | **100.** | **99.9** | **100.** |
| | Expression | 49.7 | 64.5 | 54.7 | 82.2 | 53.1 | 81.8 | 54.8 | 83.1 | **70.8** | **97.9** |
| | Average | 68.2 | 82.8 | 83.3 | 94.1 | 78.3 | 93.8 | 84.2 | 94.4 | **90.2** | **99.3** |
| CelebA | Hair color | 53.2 | 84.1 | 50.1 | 84.5 | 50.6 | 84.5 | 50.2 | 85.4 | **58.6** | **88.3** |
| | Hair style | 51.0 | 79.7 | 51.4 | 81.6 | 52.1 | 77.7 | 51.8 | 82.5 | **55.5** | **83.7** |
| | Beard style | 53.9 | 91.4 | 61.0 | 91.5 | 63.1 | 89.4 | 61.0 | 91.2 | **71.7** | **95.5** |
| | Average | 52.7 | 85.0 | 54.2 | 85.9 | 55.2 | 82.0 | 54.3 | 86.4 | **61.9** | **89.2** |

segmenting objects at fine granularity. Accordingly, we conducted an evaluation of PatchCore (Roth et al., 2022), a state-of-the-art anomaly segmentation method, alongside our selected models designed for object-level detection, using well-established anomaly detection benchmarks: MvTec (Bergmann et al., 2019) and CIFAR-10 (Krizhevsky et al., 2009). The results of this evaluation, presented in Tab. 4, highlight the varied performance of these representations. PatchCore's representations prove highly effective in identifying fine-grained anomalies, such as those found in MvTec. However, they fail when confronted with global anomalies, particularly those related to object types, as observed in CIFAR-10. Conversely, the other baseline models underperform when dealing with fine-grained anomalies. This performance difference shows that representations must align with the specific attributes and characteristics of the anomalies under consideration.

**Out-of-distribution (OOD) detection.** In OOD, the user selects a specific factor, such as an object class, that is considered crucial for identifying anomalies. The values of this factor in the normal data are labeled. In this approach, a sample exhibiting an unusual value of the specified attribute, such as belonging to a novel class, is labeled as anomalous. It is important to note that samples with highly unusual values in other attributes may not be classified as anomalous in this paradigm. Mathematically, this corresponds to choosing a representation with a $d = 1$ dimensionality. By selecting a single attribute as the basis for anomaly detection, the method achieves high accuracy if the guidance is correct and the chosen attribute is indeed the relevant one for identifying anomalies. However, if the user's guidance attribute does not align with the true relevant attribute, the insufficiency of the representation significantly reduces the accuracy of the anomaly detection process. We provide empirical support for this claim in Tab. 3. We see that relying on an irrelevant attribute for guidance in OOD detection results in significantly reduced accuracy (see App. C for details). This highlights the "no free lunch" aspect of anomaly detection representations.

**Outlier exposure (OE).** This setting specifies a set of training samples that are assumed to be similar to anomalies. The anomaly detection task then becomes supervised, similar to our linear probing setting. Unlike density estimation approaches, OE methods are not penalized for over-expressivity in their representations, as the method can learn to ignore nuisance factors. On the other hand, OE methods heavily rely on the similarity between the simulated anomalies (OE data) and the true anomalies that the model will encounter during testing. When the OE data accurately captures the characteristics of the anomalies, model performance is improved. However, if the OE data is misspecified or does not adequately represent the true anomalies, model performance is reduced.

**Nuisance disentangled representations.** This class of methods (e.g., Cohen et al. (2022), Zhang & Ranganath (2023)) directly reduces over-expressivity by explicitly removing factors of variation which the user specifies as a nuisance. When the guidance is correct, the reduced over-expressivity results in improved performance. As in the SSL case, when a relevant attribute is labeled as a nuisance, the representation runs the risk of being insufficient.

Table 3: OOD multi-value anomaly detection performance (mean ROC-AUC %) when guidance attribute is irrelevant. $k$NN best in **bold black** and linear probing best in **bold blue**.

| | Method | | OOD with irrelevant attribute | | OOD | |
|---|---|---|---|---|---|---|
| | Relevant attribute | Guidance attribute | $k$NN | Lin. | $k$NN | Lin. |
| Cars3D | Elevation | Azimuth | 55.6 | 78.7 | **69.1** | **99.2** |
| | Azimuth | Object | 70.3 | 95.3 | **84.1** | **99.9** |
| | Object | Azimuth | 67.1 | 98.0 | **96.8** | **99.9** |
| | Average | | 64.3 | 90.7 | **83.3** | **99.7** |
| RaFD | Angle | Expression | 99.8 | **100.** | 100. | **100.** |
| | Identity | Angle | 72.8 | **100.** | 99.9 | **100.** |
| | Expression | Identity | 55.3 | 84.5 | **70.8** | **97.9** |
| | Average | | 76.0 | 94.8 | **90.2** | **99.3** |
| CelebA | Hair color | Beard style | 54.2 | 84.9 | **58.6** | **88.3** |
| | Hair style | Hair color | 52.4 | 80.0 | **55.5** | **83.7** |
| | Beard style | Hair style | 64.9 | 90.4 | **71.7** | **95.5** |
| | Average | | 57.2 | 85.1 | **61.9** | **89.2** |

Table 4: Anomaly detection performance (mean ROC-AUC %) over standard benchmarks.

| Dataset | PatchCore | CSI | DN2 | PANDA | MSAD |
|---|---|---|---|---|---|
| CIFAR-10 | 74.9 | 94.3 | 92.5 | 96.2 | **97.5** |
| MvTec | **99.1** | 63.6 | 86.5 | 86.5 | 87.2 |

## 6 LIMITATIONS

**Beyond the toy model.** The simplicity of our toy model allowed us to derive fundamental insights, but real-world normal and anomalous data are often governed by more complex probability distributions and models. We hypothesize that the "no free lunch" principle remains valid for more complex distributions. This investigation is left for future work.

**Weak and few-shot supervision.** One way to avoid the over-expressivity problem is to use labeled anomalies. The labels can come in different forms, e.g., labeling just a few samples or labeling bags of samples. These techniques can be analyzed through the framework of supervised learning instead of through our framework.

**Guidance in the real world.** Designing practical guidance methods for learning representations in real-world scenarios remains an open question. Future research must take into account factors such as the availability and reliability of prior knowledge and the ease of providing guidance.

**Non density estimation methods.** We have not analyzed methods here such as auto-encoders (Venkataramanan et al., 2020) or GAN-based methods (Akcay et al., 2019). This is justified as density estimation methods (Reiss et al., 2021; Sohn et al., 2020; Roth et al., 2022) typically perform better than the other techniques. We leave an analysis of the "no free lunch" principle for other anomaly detection frameworks for future work.

## 7 CONCLUSION

In this paper, we elucidate the performance degradation of anomaly detection methods due to over-expressive representations. Our investigation revealed a fundamental trade-off between representation sufficiency and over-expressivity, highlighting the limitations of solely relying on the increased scale and expressivity of deep networks. Through a novel theoretical toy model, we have demonstrated that increasing representation expressivity does not guarantee improved anomaly detection performance. Our model establishes a "no free lunch" principle for anomaly detection, stating that increased expressivity will eventually lead to performance degradation. Furthermore, we extensively investigated state-of-the-art representations and found that they often suffer from over-expressivity, leading to failure to detect various types of anomalies other than object categories.

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

# Appendix:
# Is Scale All You Need For Anomaly Detection?

## A  PROOF OF LEMMA 1

**Lemma 1.** *Let $t_d(y) \equiv \frac{e^{-\frac{1}{2}(y\sqrt{2d}+d)}}{\sqrt{2\pi}^d}$ where $y \in \mathbb{R}$ parameterizes the sensitivity of the method. For $d$ sufficiently large for the central limit theorem to hold, the false positive rate of the method converges to a constant.*

*Proof of Lemma 1.* The relation between the false positive rate (FPR) and the parameter $y$ is given by:

$$FPR(t_d) = Pr(s_d(x) < t_d \,|l(x) = 0)$$

Recall that for the normal data $\log(s_d(x)) = -\sum_{i=1}^{d} \frac{1}{2}x_i^2 - \frac{d}{2}\log(2\pi)$, and that $\log(t_d) = -\frac{1}{2}(y\sqrt{2d}+d) - \frac{d}{2}\log(2\pi)$. Therefore, the expression can be simplified as:

$$FPR(t_d) = Pr(\sum_{i=1}^{d} x_i^2 > y\sqrt{2d} + d)$$

$$= Pr(\frac{\sum_{i=1}^{d} x_i^2 - d}{\sqrt{2d}} > y) = Pr(\frac{Q_d - \mu_d}{\sigma_d} > y)$$

where we denote $Q_d = \|x\|^2 = \sum_{i=1}^{d} x_i^2$, and $\{x_i\}_{i=1}^{d}$ are independent, standard normal random variables. Therefore, $Q_d$ is distributed according to the chi-squared distribution with $d$ degrees of freedom, i.e., $Q_d \sim \chi_d^2$. For moderately large values of $d$, as is the case with deep representations as explored in Sec. 5, the central limit theorem applies, and therefore for $d >> 1$ we have $\chi_d^2 \approx \mathcal{N}(d, 2d) = \mathcal{N}(\mu_d, \sigma_d^2)$.

We may now show that in the limit for large dimensions $d_1, d_2 \in \mathbb{N}$ the relation between $y$ and the false positive rate is independent of the dimensions:

$$FPR(t_{d_1}) = Pr(\frac{Q_{d_1} - d_1}{\sqrt{2d_1}} > y\,) = Pr(\mathcal{N}(0,1) > y\,)$$

$$= Pr(\frac{Q_{d_2} - d_2}{\sqrt{2d_2}} > y) = FPR(t_{d_2})$$

The equality of both sides of the probability of a normal random variable being larger than $y$ follows from $(d, \sqrt{2d})$ is the mean and the standard deviation of the corresponding $Q_d$ function (in the central limit theorem).

This concludes the proof of the lemma. $\qquad\qquad\square$

## B  PROOF OF THEOREM 1

**Theorem 1.** *Let the population distribution of samples (D) be given by:*

$$D = (1 - \pi) \cdot \mathcal{N}(0, I) + \pi \cdot \mathcal{N}(\Delta \cdot e_1, I) \qquad (2)$$

*where $\pi$ is a Bernoulli distribution describing the probability of a given sample to be anomalous. We assume an unknown ground truth anomaly labeling function $l : D \to \{0, 1\}$. We compute the anomaly score for test samples using their likelihood with respect to the normal data distribution: $s_d(x) = \frac{e^{-\frac{1}{2}\|x\|^2}}{\sqrt{2\pi}^d}$. Samples are classified as anomalous if their likelihood $s_d(x)$ is lower than the threshold $t_d$. We prove the following statements hold:*

1. *The true positive rate converges to the false positive rate as the dimension $d$ tends to infinity.*

2. *For large values of $d$, the difference between the true positive rate and the false positive rate decays as $\frac{1}{\sqrt{d}}$.*

*Proof of Theorem 1.* We would like to describe the distance distribution of an anomalous sample $x$ to the center of the normal data distribution, $||x||^2 = Q_d^{anom}$ to calculate their likelihood. Having the distribution of their likelihood (according to the normal data model), we can calculate their probability of being below the likelihood threshold $t_d(y)$. We begin by showing that the distance distribution ($||x||^2$) of the anomalies is of a similar distribution to that of the normal data, with a modification driven by the parameter $\Delta$. By Horgan & Murphy (2013) we have that for large $d$ for the anomalous data:

$$Q_d^{anom} = ||x||^2 = (x'_1 + \Delta)^2 + \sum_{i=2}^{d} x'^2_i \sim \mathcal{N}(2d + \Delta^2, 4\Delta^2 + 2d)$$

Where $x'$ describes the anomalous data distribution around its distribution center (shifted by $\Delta$ from the origin). Namely, $x_1 = x'_1 + \Delta$, and $x_i = x'_i$ for $i \neq 1$.

Now it holds that:

$$TPR(t_d) = Pr(s_d(x) < t_d \mid l(x) = 1) = Pr(Q_d^{anom} > y\sqrt{2d} + d\,)$$

$$= \int_{y\sqrt{2d}+d}^{\infty} \frac{1}{\sqrt{2\pi \cdot (4\Delta^2 + 2d)}} \cdot e^{-\frac{1}{2}\cdot(\frac{y'-(d+\Delta^2)}{\sqrt{4\Delta^2+2d}})^2} \, dy'$$

We make the substitution $\eta = \frac{\sqrt{2d}}{\sqrt{4\Delta^2+2d}}y'$ and get:

$$\int_{y\sqrt{2d}+d}^{\infty} \frac{1}{\sqrt{2\pi \cdot (4\Delta^2 + 2d)}} \cdot e^{-\frac{1}{2}\cdot(\frac{y'-(d+\Delta^2)}{\sqrt{4\Delta^2+2d}})^2} \, dy'$$

$$= \int_{\frac{y\cdot 2d + d\sqrt{2d}}{\sqrt{4\Delta^2+2d}}}^{\infty} \frac{1}{\sqrt{2\pi \cdot 2d}} \cdot e^{-\frac{1}{2}\cdot(\frac{\eta\sqrt{\frac{4\Delta^2+2d}{2d}}-(d+\Delta^2)}{\sqrt{4\Delta^2+2d}})^2} \, d\eta$$

$$= \int_{\frac{y\cdot 2d + d\sqrt{2d}}{\sqrt{4\Delta^2+2d}}}^{\infty} \frac{1}{\sqrt{2\pi \cdot 2d}} \cdot e^{-\frac{1}{2}\cdot(\frac{\eta}{\sqrt{2d}}-\frac{(d+\Delta^2)}{\sqrt{4\Delta^2+2d}})^2} \, d\eta$$

We make another substitution $\eta' = \sqrt{2d}(\frac{\eta}{\sqrt{2d}} - \frac{d+\Delta^2}{\sqrt{4\Delta^2+2d}} + \frac{d}{\sqrt{2d}})$ and get:

$$\int_{\frac{y\cdot 2d + d\sqrt{2d}}{\sqrt{4\Delta^2+2d}} - \frac{\sqrt{2d}(d+\Delta^2)}{\sqrt{4\Delta^2+2d}}+d}^{\infty} \frac{1}{\sqrt{2\pi \cdot 2d}} \cdot e^{-\frac{1}{2}\cdot(\frac{\eta'-d}{\sqrt{2d}})^2} \, d\eta'$$

$$= \int_{\frac{y\cdot 2d - \Delta^2\sqrt{2d}}{\sqrt{4\Delta^2+2d}}+d}^{\infty} \frac{1}{\sqrt{2\pi \cdot 2d}} \cdot e^{-\frac{1}{2}\cdot(\frac{\eta'-d}{\sqrt{2d}})^2} \, d\eta'$$

$$= \int_{\frac{y\cdot 2d - \Delta^2\sqrt{2d}}{\sqrt{4\Delta^2+2d}}+d}^{y\sqrt{2d}+d} \frac{1}{\sqrt{2\pi \cdot 2d}} \cdot e^{-\frac{1}{2}\cdot(\frac{\eta'-d}{\sqrt{2d}})^2} \, d\eta' + \int_{y\sqrt{2d}+d}^{\infty} \frac{1}{\sqrt{2\pi \cdot 2d}} \cdot e^{-\frac{1}{2}\cdot(\frac{\eta'-d}{\sqrt{2d}})^2} \, d\eta'$$

$$= \int_{\frac{y\cdot 2d - \Delta^2\sqrt{2d}}{\sqrt{4\Delta^2+2d}}+d}^{y\sqrt{2d}+d} \frac{1}{\sqrt{2\pi \cdot 2d}} \cdot e^{-\frac{1}{2}\cdot(\frac{\eta'-d}{\sqrt{2d}})^2} \, d\eta' + FPR(t_d)$$

Where the last transition follows from identifying the second part of the integral with the false positive rate we used in the proof of Lemma. 1. This rate is independent of $d$ in the limit.

We now bound the first integral by taking the maximum of the integral:

$$\leq (y \cdot \sqrt{2d} - \frac{y \cdot 2d - \Delta^2 \sqrt{2d}}{\sqrt{4\Delta^2 + 2d}}) \frac{1}{\sqrt{2\pi \cdot 2d}} \cdot e^{-\frac{1}{2} \cdot (\frac{\frac{y \cdot 2d - \Delta^2 \sqrt{2d}}{\sqrt{4\Delta^2 + 2d}}}{\sqrt{2d}})^2} + FPR(t_d)$$

$$= (y \cdot \sqrt{2d} - \frac{y \cdot 2d - \Delta^2 \sqrt{2d}}{\sqrt{4\Delta^2 + 2d}}) \frac{1}{\sqrt{2\pi \cdot 2d}} \cdot e^{-\frac{1}{2} \cdot (\frac{y \cdot \sqrt{2d} - \Delta^2}{\sqrt{4\Delta^2 + 2d}})^2} + FPR(t_d)$$

Now, we note that for the exponent:

$$\lim_{d \to \infty} \frac{y \cdot \sqrt{2d} - \Delta^2}{\sqrt{4\Delta^2 + 2d}} = y \in \mathbb{R}$$

Moreover, we observe that:

$$\lim_{d \to \infty} (y \cdot \sqrt{2d} - \frac{y \cdot 2d - \Delta^2 \sqrt{2d}}{\sqrt{4\Delta^2 + 2d}})$$

$$= \lim_{d \to \infty} y(\sqrt{2d} - \frac{2d}{\sqrt{4\Delta^2 + 2d}}) + \lim_{d \to \infty} \frac{\Delta^2 \sqrt{2d}}{\sqrt{4\Delta^2 + 2d}} = 0 + \Delta^2 = \Delta^2$$

Where, we used the fact that $\lim_{d \to \infty} y(\sqrt{2d} - \frac{2d}{\sqrt{4\Delta^2 + 2d}}) = 0$ and that $\lim_{d \to \infty} \frac{\Delta^2 \sqrt{2d}}{\sqrt{4\Delta^2 + 2d}} = \Delta^2$, and therefore the limits laws applies. As a result, we get by the limits laws that:

$$\lim_{d \to \infty} (y \cdot \sqrt{2d} - \frac{y \cdot 2d - \Delta^2 \sqrt{2d}}{\sqrt{4\Delta^2 + 2d}}) \frac{1}{\sqrt{2\pi \cdot 2d}} \cdot e^{-\frac{1}{2} \cdot (\frac{y \cdot \sqrt{2d} - \Delta^2}{\sqrt{4\Delta^2 + 2d}})^2}$$

$$= \lim_{d \to \infty} (y \cdot \sqrt{2d} - \frac{y \cdot 2d - \Delta^2 \sqrt{2d}}{\sqrt{4\Delta^2 + 2d}}) \cdot \lim_{d \to \infty} \frac{1}{\sqrt{2\pi \cdot 2d}} \cdot \lim_{d \to \infty} \cdot e^{-\frac{1}{2} \cdot (\frac{y \cdot \sqrt{2d} - \Delta^2}{\sqrt{4\Delta^2 + 2d}})^2}$$

$$= \Delta^2 \cdot 0 \cdot e^{-\frac{1}{2} \cdot y^2} = 0$$

where the limit that goes to 0 is doing so at a rate of $\frac{1}{\sqrt{d}}$. This ends the proof. $\square$

## C  EXPERIMENTAL DETAILS & MORE EVALUATIONS

### C.1  DATASET DESCRIPTIONS

In our analysis, we used three distinct datasets to evaluate our anomaly detection models. The first dataset, *Cars3D* (Reed et al., 2015), comprised a collection of 17,568 3D car models, showcasing variations in elevation, rotation (azimuth), and car model (object). The second dataset, *RaFD* (Langner et al., 2010), featured 8,040 facial images capturing different individuals expressing a wide range of emotions. Each image in the RaFD dataset was annotated with emotional attributes such as happiness, anger, and sadness, along with person ID and head pose information. Lastly, we used the *CelebA* dataset (Liu et al., 2015), which contained images of celebrity faces annotated with various attributes. For the CelebA dataset, we focused on hair color, hair style, and beard style attributes during our evaluation. These attributes were introduced by merging existing attributes in the dataset. The hair color attribute included {*Black_Hair, Blond_Hair, Brown_Hair, Gray_Hair*} values, the hair style attribute included {*Bald, Bangs, Receding_Hairline, Straight_Hair, Wavy_Hair*} values, and the beard style attribute included {*Goatee, Mustache, No_Beard*} values. We utilized the CelebA dataset test set, which consisted of 39,829 images. This test set was subsequently split into a new train and test sets for our analyses. Tab. 5 provides a summary of the attributes of interest for each dataset.

### C.2  BASELINES

In our comprehensive evaluation, we investigated various established baseline methods for anomaly detection, including both self-supervised and pretrained feature adaptation approaches. Furthermore, we incorporated an out-of-distribution (OOD) method (Fort et al., 2021) that utilizes guidance on the relevant attribute. The baselines we considered encompassed a diverse set of methods. Specifically, we evaluated DN2, which utilized unadapted ImageNet pretrained ResNet features, as well as top-performing pretrained representation adaptation methods such as PANDA (Reiss et al., 2021), MSAD

Table 5: Attribute splits for our benchmarks.

| Dataset | Attributes of Interest |
|---------|------------------------|
| Cars3D  | elevation, azimuth, object |
| RaFD    | angle, identity, expression |
| CelebA  | hair color, hair style, beard style |

(Reiss & Hoshen, 2023), and pretrained DINO (Caron et al., 2021; Reiss et al., 2023). Additionally, we included CSI (Tack et al., 2020), a leading self-supervised method that learns representations from scratch. By considering this comprehensive set of baselines, we aimed to provide a thorough analysis of different approaches for anomaly detection, spanning both self-supervised and pretrained feature adaptation paradigms.

### C.3 IMPLEMENTATION DETAILS

**Dataset splits.** For each dataset and relevant attribute, we employed an 85:15 train-test split. To evaluate our unguided anomaly detection approaches, we excluded anomalous samples from the train set. Instead, we focused solely on normal samples related to the specific attribute under investigation. Default hyperparameters were used for all methods, as specified in their official implementations.

$k$**NN density estimation.** For density estimation, we utilized the faiss (Johnson et al., 2019) library's implementation of the k-nearest neighbors ($k$NN) algorithm. To balance accuracy and computational efficiency, we set $k = 10$.

**Linear probing.** To assess over-expressivity, we employed a linear probing approach. To evaluate linear probing, we augmented the training set by introducing anomalous samples to it. Specifically, we used the complete train set from the original split, incorporating its anomalous instances with attribute-specific modifications.

We trained a linear classifier to distinguish between normal and anomalous instances based on the *representations* generated by the baselines. The training employed a stochastic gradient descent (SGD) optimizer with a binary cross-entropy loss function. The training lasted for 100 epochs, with an initial learning rate of 0.01 and a momentum factor of 0.9. During testing, the trained linear classifier was applied to compute logits which served as anomaly scores. Higher logits indicate a higher likelihood of an instance being anomalous. Importantly, the same test sets were used for both the $k$NN density estimator and linear probing anomaly scoring, ensuring a fair comparison between the two approaches.

**Out-of-distbution (OOD) detection.** We implemented an OOD method for incorporating guidance on the relevant attribute. Our approach was based on the state-of-the-art OOD method (Fort et al., 2021), which focuses on fine-tuning a pretrained backbone using a multi-class classification (cross-entropy) objective by incorporating guidance through attribute annotations. During the training phase, we included attribute values as part of the input data to guide the model's learning process. By associating attribute values with normal instances, the model learns normal data patterns and characteristics. It is important to note that the model was trained exclusively on normal data, like the other baselines.

In each experimental setting, we evaluated the performance of the anomaly detection models across the first three values of each attribute within the given dataset.

### C.4 OOD DETECTION WITH IRRELEVANT ATTRIBUTES

Our evaluation provides empirical evidence to demonstrate the impact of selecting an incorrect attribute for guidance in out-of-distribution (OOD) detection. To examine this, we conducted experiments and presented the results in Tab. 3. The table clearly illustrates that relying on an irrelevant attribute for guidance in OOD detection results in a noticeable decrease in accuracy. Specifically, in our experiment, we assigned the sequential attribute as the attribute for the irrelevant attribute OOD baseline (according to the order in Tab. 5). These findings emphasize the importance

of selecting the appropriate attribute for effective OOD detection and highlight the "no free lunch" principle in anomaly detection representations.

## C.5 TRAINING RESOURCES

Training each dataset presented in this paper takes approximately 3 hours on a single NVIDIA RTX-2080 TI.

## C.6 LICENCES

**Code.** PANDA (Reiss et al., 2021) and MSAD (Reiss & Hoshen, 2023) are licensed under a SOFTWARE RESEARCH LICENSE as described here[1]. PyTorch uses a BSD-style license, as detailed in the license file[2], while faiss (Johnson et al., 2019) creates MIT licenses.

**Metrics.** For the ROC-AUC metric we use *roc_auc_score* function from scikit-learn library.

---

[1]https://github.com/talreiss/Mean-Shifted-Anomaly-Detection/blob/main/LICENSE
[2]https://github.com/pytorch/pytorch/blob/master/LICENSE

