# OpenReview forum: "Is Scale All You Need For Anomaly Detection?"
_ICLR.cc/2024/Conference — ICLR 2024 Conference Withdrawn Submission_

### Official Review · Reviewer_EGsk · 2023-10-22

**Soundness:** 2 fair
**Presentation:** 3 good
**Contribution:** 1 poor
**Rating:** 3
**Confidence:** 4

**Summary:**

The work explores how the quality of feature representations affects the anomaly detection performance for distance-based methods. It posits that increasing the dimensionality of feature representations is not necessarily beneficial for improving the detection performance, and further demonstrates that there is a trade-off between the representation dimensionality and the detection performance. A range of experiments on three selected datasets is presented to justify the argument.

**Strengths:**

- The work studies an interesting problem of the relation between feature representations and anomaly detection accuracy. It is a problem relatively less studied by the community as most researchers focus on devising new methods/algorithms.
- A toy model and a corresponding theoretical analysis are presented to analyze the need to have a trade-off between  expressiveness of feature representations and anomaly detection accuracy.
- It further provides a set of empirical evidence on three datasets for the aforementioned trade-off argument.

**Weaknesses:**

- There are many ambiguous/undefined terms in the main claims, leading to great difficulty in understanding and evaluating the contributions. For example, 1) what is the term 'scale' referred to? Do you mean the scale of data used in the model pretraining or the number of dimensions in the feature representation space? 2) what does representation 'expressivity' mean here? does the 'expressivity' refer to the amount of relevant information w.r.t. anomaly detection or other tasks (e.g., pretext classification/prediction tasks in pre-training)? If 'expressivity' refers to the amount of relevant information w.r.t. anomaly detection, then do we still have the so-called 'over-expressivity' problem? If it refers to the amount of relevant information w.r.t. a pretext task, then it's invalid to claim something like 'over-expressivity' is bad for anomaly detection.
- Since the work focuses exclusively on anomaly detectors that calculate anomaly scores on pre-trained features, rather than anomaly detection methods that train the representation learning and anomaly scoring in an end-to-end manner, the main claims/arguments like 'over-expressivity', or no free lunch/the bias-variance trade-off do no hold for many other SOTA anomaly detection methods.
- Following the above comments, since the feature representations are fixed after the feature extraction from a pre-trained model, the studied problem becomes a classic high-dimensional anomaly detection problem, in which we concern how irrelevant/relevant features affect the detection performance. Actually, the main conclusion/claim has been studied and revealed in a number of previous studies, such as *Zimek, Arthur, Erich Schubert, and Hans‐Peter Kriegel. "A survey on unsupervised outlier detection in high‐dimensional numerical data." Statistical Analysis and Data Mining: The ASA Data Science Journal 5.5 (2012): 363-387.* and many other follow-up studies of this work.
- In the experiments, the anomaly detection methods are distance/density-based approaches working on a set of pre-trained features using some sort of pre-training tasks, so it is natural that there can be irrelevant features in the obtained feature representations. The conclusions could be the opposite of the main claim in this work, if the methods used are end-to-end trained anomaly detectors such as reconstruction/one-class-classification/knowledge-distillation-based methods.
- The main argument in Sec. 5.3 "... representations must align with the specific
attributes and characteristics of the anomalies under consideration" is kind of trivial. It is a straightforward assumption made implicitly in most detection methods.

**Questions:**

Please refer to the questions raised in the above Weaknesses section.

---

> ### Author Response · Authors · 2023-11-17
>
> The term 'scale' in our context refers to the level of expressivity in the encoded representation. Specifically, as we increase the scale, more attributes are encoded in the representation. This includes both relevant and irrelevant features in the context of anomaly detection.
>
> Regarding the term 'representation expressivity,' it indeed refers to the amount of information encoded in the representation concerning anomaly detection, not just limited to pretext classification/prediction tasks in pre-training. 'Over-expressivity' describes the scenario where the representation becomes overly rich, incorporating both relevant and irrelevant features for anomaly detection. We acknowledge the potential for confusion in the terminology and will provide additional clarification in the revised manuscript to ensure a more precise understanding.
>
> From the perspective of our analysis, there is no difference between pretrained and self-supervised methods. We consider the encoder from the input to the penultimate layer as our feature extractor, whether external data were used or not is of no consequence to our analysis.
>
> The non-trivial part of the argument is that we do not know what aspects the anomalies will take, and therefore this alignment is generally impossible to guarantee. We therefore must use priors on our target anomalies, making it impossible to have a one-size fit all method for anomaly detection. The outcome is that anomaly detection methods must be quite modest about their capabilities.

---

### Official Review · Reviewer_aDHc · 2023-10-30

**Soundness:** 2 fair
**Presentation:** 3 good
**Contribution:** 2 fair
**Rating:** 5
**Confidence:** 4

**Summary:**

The paper studies if anomaly detection problems can be solved easily with expressive representations, i.e., the representations of a neural network encoder. The authors first use a toy example to show that including more irrelevant features in the model leads to a degraded anomaly detection performance and make the statement that increasing representation expressivity does not improve the anomaly detection performance. The authors also study the effect of over-expressivity on three real-world image datasets with a single-value setup and a multi-value setup. They compare the anomaly detection performance of KNN scoring with expressive representations and linear-probe scoring with expressive representations and conclude that filtering out irrelevant features in expressive representations is a critical step for anomaly detection. They also show that having attribute labels in training (e.g., OOD detection methods) helps to learn discriminative features and then improves the anomaly detection performance.

**Strengths:**

1. The paper provides a comprehensive study of related work.

2. The paper studies the question with interpretable toy examples and real-world image datasets.

3. The empirical study covers different evaluation scenarios, i.e., the single-value and multi-value classification settings. Also, by comparing with OOD detection baselines the paper demonstrates the effect of having attribute labels in training anomaly detectors.

**Weaknesses:**

1. Although the empirical study covers different test scenarios, it only compares KNN-based scoring with the Oracle linear-probe scoring (which uses labeled anomalies in training). Including more anomaly scoring methods/models would make the evaluation more convincing.

2. In my opinion, the definition of over-expressivity is vague in the paper. The expressive representations from pretrained image models definitely contain both relevant and irrelevant features for the target anomaly detection problem. However, it provides a good basis for learning an anomaly detection model. The anomaly detection model can serve as a filter of task-relevant features. In case the representations are not expressive enough (e.g., task-relevant features are missing), an anomaly detector built upon the representations cannot solve the task anymore. In my opinion, we can consider anomaly detection a downstream problem of pretrained representation models. So, thanks to the representation expressivity, the anomaly detection model can focus on filtering out irrelevant features.

**Questions:**

Please see above

---

> ### Author Response · Authors · 2023-11-17
>
> Regarding the definition of over-expressivity, we acknowledge the importance of clarity in terminology. In our context, over-expressivity refers to the incorporation of both relevant and irrelevant features in neural representations, leading to a potential degradation in anomaly detection performance. We appreciate the reviewer's perspective on the downstream nature of anomaly detection, where expressive representations serve as a foundation for building anomaly detection models. However, it is crucial to highlight the distinctive nature of our problem setting. In anomaly detection, the relevant features defining anomalies are inherently unknown to the user prior to constructing the detection algorithm. Unlike downstream tasks where task-specific features are predetermined, anomaly detection necessitates the discovery of anomalous patterns, making it challenging to filter out irrelevant features apriori. Our empirical investigation delves into this inherent challenge by demonstrating how state-of-the-art representations, despite their expressivity, can inadvertently incorporate irrelevant features, affecting the overall anomaly detection performance.

---

### Official Review · Reviewer_ML7M · 2023-11-01

**Soundness:** 1 poor
**Presentation:** 2 fair
**Contribution:** 1 poor
**Rating:** 3
**Confidence:** 4

**Summary:**

The paper tries to answer a research question: whether it is possible to improve anomaly detection simply by adding more features. It then arrives at the conclusion that it is not possible to do so.

**Strengths:**

The research question might be important and relevant if limited within a specific family of detectors, e.g., those which make assumptions of Gaussian distribution.

**Weaknesses:**

1. The paper really only looks at some trivially simple model and incorrectly lays out the scope of the paper as overly general. The authors should very carefully and scientifically define what the true scope is.


2. The biggest issue has already been pointed out by the authors in Section 6: "The simplicity of our toy model allowed us to derive fundamental insights, but real-world normal and anomalous data are often governed by more complex probability distributions and models. We hypothesize that the ”no free lunch” principle remains valid for more complex distributions." -- The analysis in the paper was carried out on a very simple model. I do not believe it is 'fundamental' (contrary to the authors) if most real data does not follow the assumptions behind it. No reason has been presented for the hypothesis. Just because a simple model has limitations, how can we infer that complex models will suffer from the same limitations? Today, where we are encountering more complex deepnet based models, how far is the analysis in this paper relevant? With the evolution of LLMs, we are realizing that adding more data and adding more parameters does go a long way. So, maybe the analysis here is really not valid.


3. Section 5.1: 'multi-value', 'single-value' -- This is very confusing naming and easy to forget which one is what. Better to name as 'multi-normal-class', 'single-normal-class',

**Questions:**

None

---

> ### Author Response · Authors · 2023-11-17
>
> While the toy model indeed has its limitations, it serves as a valuable starting point for exploring the broader implications of over-expressivity in more complex, real-world scenarios. As stated in Section 6, we recognized that real-world data often follows more intricate probability distributions and models. However, the simplicity of the toy model allowed us to derive fundamental principles that we hypothesized could extend to more complex distributions. We want to emphasize that our paper does not contend that complex models will suffer from the exact limitations of our toy model. Instead, it asserts that the trade-off between expressivity and sensitivity is a crucial consideration in anomaly detection, even as models become more sophisticated. In our extensive empirical investigation, we applied these insights to state-of-the-art anomaly detection methods, which are notably more complex than our toy model. The empirical results presented in our paper demonstrate that these state-of-the-art methods are not immune to the challenges posed by over-expressivity. We observed significant effects on their performance, highlighting the real-world relevance of the toy model's conclusions.

---

### Author Response · Authors · 2023-11-17

We thank all the reviewers for their dedicated effort. Given the status of reviews, we will revise and submit to another conference.